# Monoclonal Antibodies for Targeted Fluorescence-Guided Surgery: A Review of Applicability across Multiple Solid Tumors

**DOI:** 10.3390/cancers16051045

**Published:** 2024-03-04

**Authors:** Stefano Giuliani, Irene Paraboschi, Angus McNair, Myles Smith, Kenneth S. Rankin, Daniel S. Elson, Vinidh Paleri, Daniel Leff, Graeme Stasiuk, John Anderson

**Affiliations:** 1Wellcome/EPSRC Centre for Interventional and Surgical Sciences, University College London, London W1W 7TY, UK; 2Cancer Section, Developmental Biology and Cancer Programme, UCL Great Ormond Street Institute of Child Health, London WC1N 1EH, UK; j.anderson@ucl.ac.uk; 3Department of Specialist Neonatal and Paediatric Surgery, Great Ormond Street Hospital for Children NHS Trust, London WC1N 3JH, UK; 4Department of Biomedical and Clinical Science, University of Milano, 20157 Milan, Italy; irene.paraboschi@hotmail.com; 5National Institute for Health Research Bristol Biomedical Research Centre, Bristol Centre for Surgical Research, Population Health Sciences, Bristol Medical School, University of Bristol, Bristol BS8 2PS, UK; angus.mcnair@bristol.ac.uk; 6Department of Gastrointestinal Surgery, North Bristol NHS Trust, Bristol BS10 5NB, UK; 7The Sarcoma, Melanoma and Rare Tumours Unit, The Royal Marsden Hospital, Institute Cancer of Research, London SW3 6JJ, UK; myles.smith@icr.ac.uk; 8Translational and Clinical Research Institute, Newcastle University, Newcastle upon Tyne NE2 4HH, UK; kenneth.rankin@newcastle.ac.uk; 9North of England Bone and Soft Tissue Tumour Service, Royal Victoria Infirmary, Newcastle upon Tyne NE1 4LP, UK; 10Hamlyn Centre for Robotic Surgery, Department of Surgery and Cancer, Imperial College London, London SW7 2AZ, UK; daniel.elson@imperial.ac.uk; 11Head and Neck Unit, The Royal Marsden Hospitals, London SW3 6JJ, UK; vinidh.paleri@rmh.nhs.uk; 12Department of Surgery and Cancer, Imperial College London, London SW7 2AZ, UK; d.leff@imperial.ac.uk; 13Imaging Chemistry and Biology, School of Biomedical Engineering and Imaging Sciences, King’s College London, St Thomas’ Hospital, London SE1 7EH, UK; graeme.stasiuk@kcl.ac.uk

**Keywords:** monoclonal antibodies, targeted fluorescence-guided surgery, solid malignancies, surgical oncology, fluorophores

## Abstract

**Simple Summary:**

Due to their specificity, monoclonal antibodies have significantly impacted cancer patients’ care, becoming one of the fastest-growing classes of new drugs approved for the treatment of solid tumors. Targeted fluorescence-guided surgery is a novel technology to better visualize tumor residuals intraoperatively. It consists of a fluorescent molecular probe, that, once injected, lights up the neoplastic cells during the surgical resection. In this regard, the development of an off-the-shelf large-scale production of clinically approved, fluorescently labeled monoclonal antibodies for targeted fluorescence-guided surgery is becoming an urgent need for oncological surgeons working in this field. Our paper aims to present the current evidence on the clinical use of monoclonal antibodies targeting solid adult and pediatric cancers. Particularly, we aim to define the utility, indications, doses, and timing of administration of the most promising monoclonal antibodies to be used for targeted fluorescence-guided surgery in oncology patients. We will also define the top three monoclonal antibodies that could be used for targeted fluorescence-guided surgery in different solid cancers, to set the basis of a bank of monoclonal antibodies that can be used to deliver highly individualized and personalized surgical approaches.

**Abstract:**

This study aims to review the status of the clinical use of monoclonal antibodies (mAbs) that have completed or are in ongoing clinical trials for targeted fluorescence-guided surgery (T-FGS) for the intraoperative identification of the tumor margins of extra-hematological solid tumors. For each of them, the targeted antigen, the mAb generic/commercial name and format, and clinical indications are presented, together with utility, doses, and the timing of administration. Based on the current scientific evidence in humans, the top three mAbs that could be prepared in a GMP-compliant bank ready to be delivered for surgical purposes are proposed to speed up the translation to the operating room and produce a few readily available “off-the-shelf” injectable fluorescent probes for safer and more effective solid tumor resection.

## 1. Introduction

Due to their target specificity and immunomodulatory properties, monoclonal antibodies (mAbs) have significantly impacted cancer patients’ care, becoming one of the fastest growing groups of new drugs approved for the cure of solid tumors [1]. mAbs have been effectively used to elicit long-lasting effector responses against solid malignancies by exploiting their unique capability of directly killing cancer cells while simultaneously activating the host immune system [1]. As a form of radioimmunotherapy, mAbs have also been adopted to selectively deliver high doses of radiation to tumor cells while also reducing the exposure of the surrounding healthy tissues, thus enhancing the efficacy of standard radiotherapy while minimizing its side effects [2].

Recently, fluorescently labeled mAbs have been used in the operating theatre to help surgeons better visualize and remove solid tumors in real time. This has been a revolution in the field of targeted fluorescence-guided surgery (T-FGS). Even if the adoption of fluorescently labeled mAbs in surgical oncology is still in its infancy, its applications are growing fast, with real potential to revolutionize the surgical field [3,4,5,6,7,8,9,10]. In this regard, the development of an “off-the-shelf” large-scale production of clinically approved fluorescently labeled mAbs is becoming an urgent need for oncological surgeons working in this surgical field.

This review paper focuses on presenting clinical evidence on the use of mAbs as the method to deliver T-FGS in adult and pediatric solid cancers. We will not consider other targeted forms of T-FGS, which are not mAb-driven, and we will exclude solid hematological cancers (e.g., lymphomas).

Particularly, we aim to define the utility, indications, doses, and timing of administration of the most promising mAbs to be used for T-FGS in oncology patients. Then, we will propose the top three mAbs that could be prepared in a GMP-compliant bank ready to be delivered for intraoperative tumor visualization.

## 2. Fluorescence-Guided Surgery (FGS): Non-Specific and Monoclonal Antibody (mAb)-Targeted

FGS is an emerging imaging tool that permits surgeons to identify healthy and pathological tissues, in real time, by intravenously delivering non-specific fluorophores (e.g., indocyanine green (ICG), 5-aminolevulinic acid, or fluorescein sodium) or fluorescently labeled molecules with a particular tissue or tumor tropism (e.g., mAbs). Depending on their route of administration, systemically administered non-specific fluorescent dyes can be used to help surgeons identify blood, bile, and lymphatic vessels during surgery, bridging the gap existing between preoperative anatomical imaging and patient-specific surgical findings [11,12].

In the surgical oncology field, non-specific dyes could be employed to distinguish neoplastic lesions from the surrounding normal tissues, by exploiting the enhanced permeability and retention (EPR) effects caused by the leaky nature of tumor vessels and the compromised lymphatic drainage [3,4,5,6,7,8,9]. Unfortunately, non-specific dyes are often unable to cause tumors from a diverse range of cancer types to fluoresce effectively as the EPR effect is not homogenous or reliable.

Besides non-specific fluorescent dyes, fluorophores have been conjugated to mAbs to deliver T-FGS, thus enhancing the intraoperative visualization of cancer at the tumor margins, loco-regional lymph nodes, and occult disease with high resolution and deep tissue penetration. By binding to a specific site of the tumor antigen (i.e., epitope), fluorescently labeled mAbs can be administered a few days before surgery, and then visualized using new-generation near-infrared (NIR) camera systems for endoscopic and open-field procedures [3,4,5,6,7,8,9].

Building on the widespread use of non-specific FGS and ICG, biomedical device companies have developed several NIR fluorescence platforms, which can be used to detect mAb-based FGS [3,4,5,8]. Briefly, the fluorophore-conjugated mAb needs to be excited at a specific wavelength to produce an invisible light, which will be collected at a set wavelength range to be transformed into an image. For surgical indications, mAbs have been conjugated to NIR dyes (e.g., IRDye800CW; wavelengths: 650–900 nm), which provide low absorption in healthy tissue, high tumor-to-background-ratio (TBR) and tissue penetration, with low interference from intrinsic fluorescence, especially when compared to the visible light spectrum (wavelengths: 400–600 nm). By doing so, the non-specific tissue background can be minimized while the tumor cells can be illuminated in real-time helping surgeons with better visualization [3,4,5,6,7,8,9].

## 3. Monoclonal Antibodies (mAbs) Clinically Approved for the Treatment of Extra-Hematological Solid Malignancies under Evaluation in Clinical Trials for T-FGS Purposes

In this section, mAbs clinically approved for the treatment of extra-hematological solid malignancies are presented. For each of them, we will report the antigen targeted, mAb generic/commercial name and format, and clinical indication (Table 1). Then, we will present completed or ongoing human clinical trials using mAbs for T-FGS with a focus on indications, doses, the timing of administration, and safety profile (Table 2).
**mAbs targeting the epidermal growth factor receptor (EGFR)**

The EGFR (c-ErbB1, HER1) is a 170 kDa transmembrane glycoprotein belonging to the epidermal growth factor receptor (ErbB) family [13,14].

The members of this family of type I receptor tyrosine kinases, which also include c-ErbB2 (HER2, neu in rodents), c-ErbB3 (HER3), and c-ErbB4 (HER4), share similarities in their structure (i.e., an extracellular domain, a lipophilic transmembrane region, an intracellular domain containing tyrosine kinase, and a carboxy-terminal region) and function [13,14].

Despite being a monomer in its inactive form, when it is bounded by its ligands (e.g., EGF, TNFα), EGFR forms homodimers or heterodimers with another member of its family of receptors and activates its intracellular tyrosine kinase region. This results in its autophosphorylation and in the initiation of a cascade of intracellular events, which regulate tumor cell differentiation, proliferation, angiogenesis, and migration [13,14].

Several alterations in EGFR are associated with the growth and spread of different solid tumors in humans. Hence, this surface antigen has been extensively studied over time together with the effects of anti-EGFR mAbs on EGFR-positive tumors [13,14].

Since anti-EGFR mAbs are bound to EGFR with a higher affinity compared with its ligands, they inhibit the subsequent activation of the tyrosine kinase-mediated cascade of intracellular events [13,14].

**Cetuximab (Erbitux)** is a recombinant human/mouse chimeric IgG1 mAb that binds to the extracellular domain of the human EGFR and, therefore, competitively inhibits EGF and other ligand binding. The inhibition of cell proliferation, the induction of apoptosis, and reduced matrix metalloproteinase and vascular endothelial growth factor (VEGF) production are all results of the binding of Cetuximab to its receptor. This mAb is currently approved for the treatment of recurrent squamous-cell carcinoma of the head and neck, K-Ras wild-type EGFR-expressing colorectal cancer, and BRAF V600E mutation-positive metastatic colorectal cancer. The safety and tumor specificity of Cetuximab-IRDye800CW as a surgical navigation tool have been successfully demonstrated in multiple in vitro and in vivo models of human cancers, showing limited toxicity. Moreover, several early-phase clinical trials are in progress to test the safety, efficacy, reactions, best dosage, formulation method, and efficacy of its administration. Table 2 lists the solid tumors currently under investigation for T-FGS applications of Cetuximab-IRDye800CW, including esophageal cancer (ClinicalTrials.gov NCT04161560), head and neck cancer (ClinicalTrials.gov NCT03134846, NCT01987375), pancreatic adenocarcinoma (ClinicalTrials.gov NCT02736578), brain neoplasm (ClinicalTrials.gov NCT02855086), rectal cancer (ClinicalTrials.gov NCT04638036), and penile cancer (ClinicalTrials.gov NCT05376202).

In 2015, Rosenthal et al. [15] published a dose escalation study of Cetuximab- IRDye800CW enrolling 12 patients undergoing surgery for squamous-cell carcinoma of the head and neck. This report represents a milestone in the application of tumor biology-based FGS because it proved for the first time that commercially available mAb could be labeled with fluorophores and safely administered to patients in order to identify in real-time solid tumors with sharp demarcation and deep resolution. According to the study design, the escalating doses were based on the therapeutic dose of Cetuximab (250 mg/m^2^): 2.5 mg/m^2^, 25.0 mg/m^2^, and 62.5 mg/m^2^, corresponding to the 1%, 10%, and 25% of the therapeutic dose. Wide-field NIR imaging was performed after infusion on days 0, 1, and on the day of surgical resection. While no severe side effects were reported, an average TBR of 5.2 in the highest dose range helped successfully differentiate the tumor from the surrounding healthy tissue during surgery.

Three years later, Miller et al. [16] used Cetuximab conjugated with IRDye800CW for intraoperative imaging during glioblastoma resection. Two doses (50 and 100 mg) were administered 2–5 days before surgery to identify the optimal TBR. It is worth noting that a 100 mg loading dose of unlabeled Cetuximab was infused before the fluorescent agent to differentiate potential adverse reactions and to saturate tissues highly expressing EGFR (e.g., liver). Interestingly, the administered dose correlated with the TBR (mean: 3.6 vs. 4.3) and the smallest detectable tumor volume (70 mg vs. 10 mg), and so did the specificity (66.3% vs. 69.8%) and sensitivity for viable tumor tissue (73.0% vs. 98.2%). With the limitation of a small number of patients included, this study represents a pivotal proof-of-principle of the feasibility, safety, and capacity of mAb-based imaging to traverse the blood–brain barrier for contrast-enhancing glioblastomas.

**Panitumumab (Vectibix)** is a fully human IgG2 anti-EGFR mAb, currently used for the treatment of patients with wild-type RAS (defined as wild-type in both KRAS and NRAS as determined by an FDA-approved test for this use) metastatic colorectal cancer.

Panitumumab-IRDye800CW has been employed for FGS in patients with head and neck cancer (ClinicalTrials.gov NCT04511078, NCT03405142, NCT02415881, NCT03733210), lung cancer (ClinicalTrials.gov NCT03582124), brain tumor (ClinicalTrials.gov NCT04085887, NCT03510208), and pancreatic adenocarcinoma (ClinicalTrials.gov NCT03384238).

In 2018, Gao et al. [17] published the results of a Phase I clinical trial enrolling 15 patients with biopsy-confirmed head and neck squamous-cell carcinoma undergoing standard surgery.

One to five days before surgery, 100 mg of unlabeled Panitumumab were administered to patients as a loading dose to limit background. In case of no drug reaction, a dose of 0.06 mg/kg Panitumumab-IRDye800CW (1/100 of one therapeutic dose), 0.5 mg/kg (1/12 of one therapeutic dose), or 1 mg/kg (1/6 of one therapeutic dose) was infused. The results of this study together with those of similar mAb-based imaging studies suggested that Panitumumab-IRDye800CW has toxicity and pharmacodynamic profiles like the unconjugated mAb, and that its safety profile can be exploited to design safe and effective fluorescent imaging probes at doses up to 25% of the therapeutic one.

Moreover, by histologically correlating the tumor location with fluorescence intensities, the authors reported a high TBR, sensitivity, specificity, positive predictive value (PPV), and negative predictive value (NPV) of the tested optical imaging agent at both doses (0.5 mg/kg: 9.96 ± 1.30; 80.0%; 94.8%, 42.7%, and 96.5%, respectively; 1 mg/kg: 8.85 ± 1.26; 83.7%; 72.1%; 20.6%, and 99.5%; respectively) that significantly aided in the real-time detection of surgical resection margins of head and neck squamous-cell carcinomas.

More recently, Zhou et al. [18] investigated the use of Panitumumab-IRDye800CW, administered at subtherapeutic doses, as a safe and effective imaging agent for high-grade gliomas. According to the study design, 11 patients affected by contrast-enhancing high-grade gliomas were administered Panitumumab-IRDye800CW at two doses (low: 50 mg; high: 100 mg) 1–5 days before surgery. The authors concluded that Panitumumab-IRDye800CW was safe at both doses and provided excellent intraoperative contrast in a dose-dependent way (smallest detectable tissue weights: 3 mg vs. 66 mm), with no difference in the observed NIR fluorescence in tumor tissue resected among the three imaging windows. Although viable tumor tissue could be visualized with PPV and NPV above 85% at both doses, the highest dose provided a superior AUC compared to 50 mg (0.90 vs. 0.85), which was thus more appropriate for real-time surgical imaging.
**mAbs targeting the vascular endothelial growth factor (VEGF)**

Angiogenesis is a complex phenomenon, involving several series of events, including the outgrowth of post-capillary vessels from pre-existing venules, vasodilation, increased permeability, and the migration of endothelial progenitor cells. Besides the tightly regulated process occurring during the normal growth of an organism and the tissue repair occurring during wound healing, solid malignancies also upregulate angiogenetic processes to induce nutrient delivery and tumor growth [19,20].

In this regard, the protein family of VEGF has been demonstrated to play major roles in physiological and pathological angiogenesis by providing signaling pathways and cascades of events crucial for endothelial cell division and migration. The VEGF family consists of six secreted proteins including VEGF-A (also known as VEGF), VEGF-B, VEGF-C, VEGF-D, VEGF-E, and placenta-induced growth factor (PDGF), characterized by eight conserved cysteines and functions as a homodimer structure. These ligands bind to three different, but structurally related, VEGF-receptor (VEGFR) tyrosine kinases, essential for hematopoietic, vascular endothelial, and lymphatic endothelial cell development [19,20].

**Bevacizumab (Avastin)** is a recombinant humanized anti-VEGF IgG1 mAb that contains human framework regions and murine complementarity-determining regions. Bevacizumab binds VEGF and interferes with the interaction of VEGF with its receptors (Flt-1 and KDR) on the surface of endothelial cells, preventing angiogenetic events.

It has been approved for patients affected with metastatic colorectal cancers, unresectable, locally advanced, recurrent, or metastatic non-squamous non-small-cell lung cancers, recurrent glioblastomas, metastatic renal-cell carcinomas, persistent, recurrent, or metastatic cervical cancers, epithelial ovarian, fallopian tube, or primary peritoneal cancers, and unresectable or metastatic hepatocellular carcinoma.

Different treatment protocols have been described according to the different histological subtypes, with specific guidelines and recommendations for drug infusion, dosage calculations, ortive treatment, the monitoring of vital parameters, and criteria for dose changes.

Bevacizumab-IRDye800CW has been leveraged for delivering precision surgery in several clinical trials enrolling patients with adenomatous polyposis coli (ClinicalTrials.gov NCT02113202), rectal cancer (ClinicalTrials.gov NCT01972373), breast cancer (ClinicalTrials.gov NCT01508572), soft-tissue sarcoma (ClinicalTrials.gov NCT03913806), Barrett esophagus (ClinicalTrials.gov NCT03877601), esophageal cancer (ClinicalTrials.gov NCT02129933, NCT03558724), hilar cholangiocarcinoma (ClinicalTrials.gov NCT03620292), inverted papilloma (ClinicalTrials.gov NCT03925285), pancreatic cancer (ClinicalTrials.gov NCT02743975), and pituitary tumor (ClinicalTrials.gov NCT04212793).

In 2016, Harlaar et al. [21] ran a single-center feasibility study to prove the safety of molecular FGS using Bevacizumab-IRDye800CW in seven patients with colorectal peritoneal metastases scheduled for cytoreductive surgery and hyperthermic intraperitoneal chemotherapy. No serious side effects related to its administration were reported in any of the patients enrolled receiving 4.5 mg (1 mg/mL) of the NIR fluorescent tracer 2 days before surgery. Of interest, fluorescence was seen intraoperatively in all patients, and in two patients, additional tumor tissues that were initially missed by the surgeons were detected by FGS itself. Although molecular FGS was associated with more false-positive results than expected initially (in fact, tumor tissue was detected in 27 of 51 fluorescent areas, 53%), no residual disease was found in the absence of fluorescence.

A year later, Lamberts et al. [22] published a proof-of-principle study describing the safety of 4.5 mg Bevacizumab-IRDye800CW in 21 patients affected with primary invasive breast cancer eligible for primary surgery. Interestingly, none had side effects, and even if in situ intraoperative tumor margin detection was not possible (due to the microdose of the tracer administered) the ex vivo imaging proved to correlate well with VEGF-A quantification and microscopic analyses of the tumor site of targeting. Further studies adopting higher tracer doses, albeit still well below the therapeutic dosing scheme of Bevacizumab, were therefore warranted for intraoperative in situ imaging purposes.

More recently, in 2020, J. de Jongh et al. [23] demonstrated the potential of back-table FGS for the evaluation in the surgical theatre of the margin of locally advanced rectal cancers following the administration of 4.5 mg of Bevacizumab-800CW 2–3 days before surgery. The technique itself proved to be safe and feasible and showed huge potential for guiding intraoperative decision-making with high sensitivity.

In the same year, Tjalma et al. [24] evaluated the role of quantitative fluorescence endoscopy targeting VEGF A to detect residual tumors after neoadjuvant chemoradiotherapy in a first-in-human pilot by enrolling 25 patients with locally advanced rectal cancer who received 4.5 mg of Bevacizumab-800CW intravenously 2–3 days prior to endoscopy. Their results demonstrated that this image-guided technique aided clinical response assessment in the patients by showing significantly higher fluorescent signals in tumors compared with healthy rectal tissue and fibrosis and by improving the prediction of final pathology results in 16% of patients compared with standard MRI and white-light endoscopy. Further studies involving more patients are warranted to realize this strategy.
**mAbs targeting carbonic anhydrase IX (CAIX)**

Carbonic anhydrase IX (CAIX) is a zinc metalloprotein enzyme. CAIX belongs to the carbonic anhydrase family. Under conditions associated with low pH, CAIX is triggered by the hypoxia-inducible factor 1-α (HIF1-α) and is highly upregulated in environments characterized by a low pressure of oxygen. By catalyzing the interconversion of CO_2_ and HCO_3_^−^ to maintain intracellular pH homeostasis, the overexpression of CAIX contributes to cancer cell survival. Not only does CAIX modulate the microenvironment, but it also has a critical role in mediating tumor survival, spreading, and metastasis [25]. Hence, CAIX has become an interesting target not only for cancer diagnosis but also for treatment. Its expression is increased in a wide variety of solid tumors (e.g., renal-cell carcinoma, brain, bladder, cervical, head and neck, breast, lung, sarcoma, and kidney cancers). Conversely, its expression in normal healthy tissues is usually low [26]. It is worth noting that the Von Hippel–Lindau mutation in clear-cell renal-cell carcinomas causes HIF1-α overexpression on the cell surface in this group of tumors [27].

**Girentuximab (Rencarex)** is a chimeric IgG1 mAb that binds CAIX and triggers antibody-dependent cell-mediated cytotoxicity (ADCC) by activating natural killer cells.

As an adjuvant treatment for high-risk clear-cell renal-cell carcinoma patients, its development as an unconjugated antibody was suspended because it showed no clinical benefit in a Phase III trial [28]. However, as a radioimmunoconjugate (i.e., 89Zr-Girentuximab), its safety and tolerance were proven in all patients, with a mean administered activity of 36.7 ± 1.1 MBq (range: 34.2–38.34 MBq) in a Phase I clinical trial [29].

In 2018, Hekman et al. [30] performed a Phase I dose-escalation study investigating the use of [111In] In-DOTA-Girentuximab-IRDye800CW in 15 patients with primary renal mass scheduled for surgery.

Four days after the administration of [111In] In-DOTA-Girentuximab-IRDye800CW (5, 10, 30, or 50 mg), SPECT/CT imaging was performed, whereas surgery was performed seven days after the infusion by using a gamma probe and NIR fluorescence camera. All CAIX-expressing tumors were visualized with SPECT/CT, and no uptake was observed in CAIX-negative tumors, proving the high sensitivity of the probe. Interestingly, an excellent concordance was also observed between the SPECT/CT and the NIR imaging. With a mean TBR of 2.5 at all protein doses, all clear-cell renal-cell carcinoma could be localized with gamma probe measurements during surgery. In summary, this Phase I study proved the safety of the [111In]In-DOTA-Girentuximab-IRDye800CW for targeting CAIX during clear-cell renal-cell carcinoma resection and it could be easily adopted for intraoperative guidance and decision-making.
**mAbs targeting carcinoembryonic antigen (CEA)**

Carcinoembryonic antigen (CEA) is a non-specific serum biomarker that is elevated in various malignancies, and it is highly expressed on the surface of colorectal cancer cells in more than 90% of all cases with a significant antigenic density. Conversely, its expression is 60 times lower on average in healthy tissue [31].

**Labetuzumab (CEA-CIDE)** is a humanized IgG1 mAb that has a high specificity to CEA. It has been deeply studied as a therapeutic drug, radiotracer, and antibody–drug conjugate for the treatment of several tumors.

A Phase I clinical trial was designed for the accurate preoperative evaluation and subsequent intraoperative identification of all tumor deposits in patients with colorectal peritoneal metastases undergoing cytoreductive surgery by using a dual-labeled version of Labetuzumab, the [111In]In-DOTA-Labetuzumab-IRDye800CW probe [31].

The results of the trial showed that [111In]In-DOTA-Labetuzumab-IRDye800CW was safe and enabled sensitive pre- and intraoperative imaging in patients who received 10 or 50 mg of the dual-labeled anti-CEA antibody conjugate. Interestingly, the preoperative imaging study showed previously undetected lymph node metastases in one case, while the intraoperative fluorescence signal allowed the identification of four previously undetected metastases in two cases. Moreover, in three patients, this multimodal imaging altered the decision-making process and the clinical and surgical strategy. A dose-expansion study was therefore warranted to further investigate the diagnostic accuracy and the potential implications of this dual-labeled anti-CEA antibody conjugate for patients with peritoneal metastases derived from colorectal cancer.

## 4. Monoclonal Antibodies (mAbs) Clinically Approved for the Treatment of Extra-Hematological Solid Malignancies Not Yet under Evaluation in Clinical Trials for T-FGS Purposes

In this section, mAbs clinically approved for the treatment of extra-hematological solid malignancies but not yet under clinical trial for T-FGS purposes are presented. For each of them, we will report the antigen targeted, mAb generic/commercial name and format, and clinical indication (Table 1). The aim is to stimulate future research and company developments in the field of T-FGS.
**mAbs targeting the epidermal growth factor receptor (EGFR)**

**Necitumumab (Portazza)** is an anti-EGFR recombinant human mAb of the IgG1 kappa isotype. With gemcitabine and cisplatin, it is used as a first-line treatment for patients affected by metastatic squamous non-small-cell lung cancer.
**mAbs targeting the epidermal growth factor receptor 2 (HER2)**

HER2 is a transmembrane receptor protein of 185 kDa, structurally related to the EGFR. In HER2-overexpressing tumors, an aberrant HER2 activity results in receptor dimerization (e.g., HER2/HER3), triggering complex intracellular signaling pathways that modulate cancer cell survival, proliferation, mobility, and invasiveness [32].

**Trastuzumab (Herceptin)** is a recombinant DNA-derived humanized IgG1 mAb that selectively binds with high affinity to the extracellular domain of HER2.

Trastuzumab is currently employed as an adjuvant treatment of HER2-overexpressing node-positive or node-negative (ER/PR negative or with one high-risk feature) breast cancers, in combination with paclitaxel for the first-line treatment of HER2-overexpressing metastatic breast cancers, and as a single agent for the treatment of HER2-overexpressing breast cancer in patients who have received one or more chemotherapy regimens for metastatic disease. Trastuzumab is also the basics of two approved antibody–drug conjugates (i.e., Ado-Trastuzumab Emtansine and Trastuzumab-Deruxtecan) for patients with HER2-positive metastatic breast cancer whose disease progresses after treatment with a combination of anti-HER2 antibodies and a taxane.

In combination with cisplatin and capecitabine or 5-fluorouracil, it is also used for the treatment of patients with HER2-overexpressing metastatic gastric or gastro-esophageal junction adenocarcinoma who have not received prior treatment for metastatic disease.

**Pertuzumab (Perjeta)** is a recombinant humanized mAb that binds the extracellular dimerization domain (subdomain II) of HER2. By doing so, it blocks the ligand-dependent heterodimerization of HER2 with other HER family members (e.g., EGFR, HER3, and HER4). By inhibiting ligand-initiated intracellular cascades through two major signal pathways (i.e., mitogen-activated protein [MAP] kinase and phosphoinositide 3-kinase [PI3K]), Pertuzumab results in cell growth arrest and apoptosis. Moreover, Pertuzumab is a mediator of ADCC.

In 2017, the FDA granted regular approval to Pertuzumab for its use (i) in association with Trastuzumab and Docetaxel, for the treatment of HER2-positive metastatic breast cancers that had not been treated with prior anti-HER2 therapy or chemotherapy for metastatic disease; (ii) in association with Trastuzumab and chemotherapy, as a neoadjuvant treatment of HER2-positive, locally advanced, inflammatory, or early-stage breast cancers (either greater than 2 cm in diameter or node-positive); and (iii) as an adjuvant treatment of HER2-positive early-stage breast cancers at a high risk of recurrence.
**mAbs targeting the vascular endothelial growth factor receptor 2 (VEGFR2)**

VEGFR2 is a type III transmembrane kinase receptor composed of 1356 amino acids. It is made of an extracellular region consisting of seven immunoglobulin (Ig)-like domains, a short transmembrane domain, and an intracellular region containing a tyrosine kinase domain, split by a 70-amino-acid insert [19,20].

In adults, VEGFR-2 is mostly expressed on vascular endothelial cells and mediates several physiological and pathological processes driven by VEGF-A, including cell survival, proliferation, migration, and increased vascular permeability [19,20].

**Ramucirumab (Cyramza)** is a recombinant human IgG1 mAb targeting the extracellular domain of VEGFR2 and blocking the binding of VEGFR ligands, VEGF-A, VEGF-C, and VEGF-D. As a consequence, Ramucirumab inhibits the ligand-stimulated activation of VEGFR2, thereby inhibiting the ligand-induced proliferation and migration of human endothelial cells.

Ramucirumab is currently approved for the treatment of patients with advanced or metastatic gastric or gastro-esophageal junction adenocarcinomas, metastatic non-small-cell lung cancers, metastatic colorectal cancers, or hepatocellular carcinomas.
**mAbs targeting platelet-derived growth factor receptor α (PDGFRα)**

PDGFR-α is a receptor tyrosine kinase expressed on cells of mesenchymal origin [33,34]. Signaling through this receptor plays a role in cell proliferation, migration, mesenchymal stem cell differentiation, and VEGF-mediated angiogenesis [33,34].

The overexpression and activating mutations of PDGFRα in tumor cells have been demonstrated to contribute to tumor growth, metastasis, and the establishment of a tumor-supporting microenvironment. In this regard, a range of tyrosine kinase inhibitors with activity against PDGFRα have been investigated, aiming to reduce the proliferation and spreading of numerous solid cancers including sarcomas [33,34].

**Olaratumab (Lartruvo)** is a recombinant human IgG1 mAb that binds specifically to PDGFRα. It is used, in association with doxorubicin, for the treatment of adult patients with soft-tissue sarcoma with a histologic subtype for which an anthracycline-containing regimen is indicated and which is not amenable to curative treatment with radiotherapy or surgery.
**mAbs targeting programmed death ligand-1 (PD-L1)**

PD-L1 is expressed in several cell types: B cells, T cells, dendritic cells (DC), macrophages, mast cells, placenta, vascular endothelium, pancreatic islet cells, muscle, hepatocytes, epithelium, and mesenchymal stem cells. In pathological tumor cells, PD-L1 is overexpressed to escape immunologic surveillance and facilitate cancer growth. Several solid malignancies including melanoma, renal-cell carcinoma, non-small-cell lung cancer, thymoma, ovarian, and colorectal cancer express PD-L1 to create an immunosuppressive tumor microenvironment and escape T-cell cytolysis. Recently, several anti-PD-L1 mAbs have proven their safety and efficacy in causing durable antitumor immune effects with low toxicity for solid tumor types [35,36].

**Atezolizumab (Tecentriq)** is an Fc-engineered, humanized, non-glycosylated IgG1 kappa PD-L1-blocking mAb. It is clinically approved for the treatment of patients with non-small-cell lung cancers, small-cell lung cancers, hepatocellular carcinomas, melanomas, and alveolar soft part sarcomas.

**Avelumab (Bavencio)** is a human IgG1 lambda PD-L1 blocking mAb approved for the treatment of adults and pediatric patients 12 years and older with metastatic Merkel cell carcinomas.

**Durvalumab (Imfinzi)** is a human IgG1 kappa mAb that binds to PD-L1 and blocks the interaction of PD-L1 with PD-1 and CD80. It is indicated for the treatment of adult patients with unresectable, stage III non-small-cell lung cancer or with metastatic non-small-cell lung cancer with no sensitizing EGFR mutations or anaplastic lymphoma kinase (ALK) genomic tumor aberrations. Moreover, it is approved as a first-line treatment for adult patients with extensive-stage small-cell lung cancers or with locally advanced or metastatic biliary tract cancer. Additional indications are represented by unresectable hepatocellular carcinomas.
**mAbs targeting the disialoganglioside GD2**

Gangliosides (e.g., GM3, GM2, GM1, and GD1) are carbohydrate-containing sphingolipids highly expressed in healthy tissues. This explains why most subtypes are unsuitable targets for cancer therapy. On the contrary, the disialoganglioside GD2 is considered a tumor-associated antigen thanks to its minimal expression in healthy human tissues (e.g., lymphocytes, mesenchymal stem cells, dermal melanocytes, central nervous system, and peripheral sensory nerve fibers,) and good expression in several tumors (e.g., neuroblastomas, melanomas, retinoblastomas, and several Ewing sarcomas), and, hence, valued as a primary target for cancer immunotherapy [37,38,39] (see Figure 1). Not only its peculiar expression profile on solid tumors, but also its role in enhanced tumor cell proliferation, growth, motility, migration, adhesion, and invasion highlight its role for clinical targeting this tumor antigen with mAbs for therapeutic approaches [37,38,40,41]. Based on all these characteristics, the disialoganglioside GD2 has been ranked 12th among 75 potential targets for anti-cancer therapy in 2009 by the U.S. National Cancer Institute [38].

**Dinutuximab (Unituxin)** is a chimeric GD2-binding mAb employed for the treatment of children with high-risk neuroblastoma who achieve at least a partial response to prior first-line multiagent multimodality therapy.

**Dinutuximab-beta (Quarziba)** is a murine-human chimeric anti-GD2 mAbs produced in Chinese hamster ovary (CHO) cells. In May 2017, the European Commission approved Dinutuximab-beta for the treatment of high-risk neuroblastoma in patients aged 12 months and above who have previously received induction chemotherapy and achieved at least a partial response followed by myeloablative therapy and stem cell transplantation. Patients with a history of relapsed or refractory neuroblastoma with or without residual disease are also potential candidates for its use.

## 5. Promising Monoclonal Antibodies (mAbs) for T-FGS Purposes Not Yet Clinically Approved for the Treatment of Extra-Hematological Solid Tumors


**mAbs targeting the epidermal growth factor receptor (EGFR)**


**Nimotuzumab (h-R3, BIOMAb EGFR, Biocon,** India; **TheraCIM**, CIMYM Biosciences, Canada; Theraloc, Oncoscience, Europe, **CIMAher**, Center of Molecular Immunology, Havana, Cuba) is a humanized anti-EGFR mAb, that was obtained in 1996 after the genetic modification of the parental murine molecule ior egf/r3 [42]. This mAb inhibits cell growth and angiogenesis, activates NK cells, induces cytotoxic T cells, and stimulates DC maturation. By hindering one of the EGFR immune-escape ways, this mAb restores MHC-I expression on tumor cells. Nimotuzumab has been deeply examined in seven clinical trials enrolling patients with inoperable head and neck tumors, concurrently with irradiation alone or irradiation plus chemotherapy [42].

According to a 2009 review [43], Nimotuzumab was approved for “squamous cell carcinoma in head and neck in India, Cuba, Argentina, Colombia, Ivory Coast, Gabon, Ukraine, Peru, and Sri Lanka; for glioma (pediatric and adult) in Cuba, Argentina, Philippines, and Ukraine; for nasopharyngeal cancer in China. It has been granted orphan drug status for glioma in the USA and for glioma and pancreatic cancer in Europe”. In April 2014, Daiichi Sankyo announced the discontinuation of a Phase III clinical trial conducted in Japan investigating Nimotuzumab for first-line therapy in patients with unresectable and locally advanced squamous-cell lung cancer based on safety issues occurring in certain patients receiving a combination of cisplatin, vinorelbine, and radiotherapy [44]. A phase I/II clinical trial evaluating Nimotuzumab-IRDye800CW as a NIR-I imaging probe for image-guided surgery is currently determining the safety, optimal dose, and imaging time for lung cancer resection (ClinicalTrials.gov NCT04459065).
**mAbs targeting B7-H3 (CD276)**

B7-H3 (CD276) is a member of the B7 family of immune checkpoint molecules, that is overexpressed in cancer cells and in activated tumor-infiltrating immune cells. B7-H3’s main function is to aid tumor cells to escape the surveillance of cytotoxic T cells and NK cells [45].

Several studies have demonstrated that B7-H3 is involved in the growth, spreading, and treatment resistance of several cancer types (e.g., lung cancer, colorectal cancer, breast cancer, prostate cancer, melanoma, gastric cancer, liver cancer, cervical cancer, and brain tumors), resulting in poor patient outcomes [46]. B7H3 has been also reported to be expressed specifically in the endothelial cells of tumors but not in normal vascular endothelium [47]. The significant difference in the expression levels of this immune checkpoint between healthy and tumor tissues can be adopted to develop targeting drugs with cancer-specific toxicity, without harm to normal tissue. B7-H3 is, therefore, a promising target for cancer therapy, including T-FGS. As a matter of fact, immunotherapies targeting B7-H3 have been developing rapidly, and many ongoing clinical trials are currently investigating the safety and efficacy profiles of this immune checkpoint molecule in cancer treatments (ClinicalTrials.gov NCT02982941, NCT02923180, NCT02381314, NCT04630769, NCT04634825, and NCT02475213).

## 6. Discussion

Several mAb-fluorophore conjugates have been effectively used for T-FGS in humans. We believe that the three most versatile ones are Cetuximab-IRDye800CW, Panitumumab-IRDye800CW, and Bevacizumab-IRDye800CW.

Among these three, examining the timing/dose of administration, safety profile, and side effects, we believe that Cetuximab-IRDye800CW has reached sufficient scientific evidence to support its role as a commercially available mAb for T-FGS in humans.

To reach the highest TBR and avoid a specific binding, Cetuximab should be administered intravenously 2 or 3 days before surgery as an unlabeled test/loading dose, followed by the fluorescently labeled one.

Although many solid tumor types have been investigated in clinical trials so far (i.e., head and neck cancer, esophageal cancer, pancreatic adenocarcinoma, brain neoplasm, rectal cancer, and penile cancer), several more can benefit from fluorescently labeled mAbs in the future to deliver highly individualized and personalized surgical approaches.

However, some key limitations of T-FGS have yet to be resolved.

Firstly, NIR-light penetration up to a few millimeters may limit the application of this modality for deep-seated lesions within solid organs (including minimal residual disease) and lymph node metastases. In this regard, more recent studies are evaluating dye with longer excitation wavelengths (i.e., NIR-II [wavelength: 1100–1350 nm], NIR-III [wavelength: 1600–1850 nm], and NIR-IV [wavelength: 2100–2300 nm] dyes) for achieving higher contrast, greater sensitivity, and improved penetration depths [48,49]. Despite the fact that no NIR-II fluorophores have been approved for clinical use in humans yet, some NIR-I dyes (e.g., ICG) have been shown to display bright emission tails over 1000 nm. This offers exciting opportunities for T-FGS [50].

Secondly, mAbs need to be given a few days before surgery, requiring an additional visit to the hospital for the patient. In this regard, new-generation small molecules have been recently developed for immunotherapy (i.e., affibodies and nanobodies), offering exciting new opportunities in the field of T-FGS. These small molecules could be adopted as theranostic tools to provide high-contrast images within a few hours after injection, overcoming the slow and low tumor uptake and the limited tumor penetration compared to traditional mAbs [51].

Thirdly, another limitation of fluorescently labeled mAbs conjugates is the quick quenching process, which means the loss of fluorescence due to a gain in stability from the electrons present in the fluorophore. Intraoperative sprayable targeted probes could overcome this problem. They will be applied to the region of interest (ROI) with a simple spray and make their clinical use simpler, more convenient, and possibly with fewer systemic side effects [3].

Finally, a thorough characterization of cancer surface antigen expression should become routine at diagnosis to decide which is the best mAb to deliver T-FGS at the time of surgical removal (precision medicine).

## 7. Conclusions

In this manuscript, the status of the clinical use of mAbs in cancer surgery was reviewed. This review aims to facilitate future research and molecular probe development in the field of T-FGS. This field is developing fast, and there is an opportunity to speed up the translation to the operating room by producing a few readily available “off-the-shelf” injectable fluorescence probes effective for several solid cancers.

## Figures and Tables

**Figure 1 cancers-16-01045-f001:**
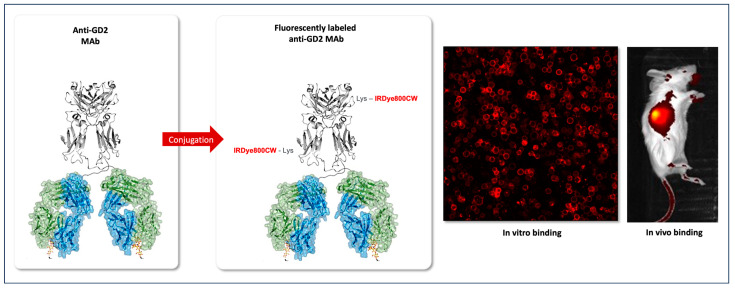
Conjugation of anti-GD2 monoclonal antibodies (mAbs) with IRDye800CW and in vitro and in vivo binding experiments on neuroblastoma cells expressing the GD2 antigen.

**Table 1 cancers-16-01045-t001:** Monoclonal antibodies (mAbs) clinically approved for the treatment of solid extra-hematological malignancies.

Antigen	Drug Generic Name	Drug Brand Name	Format	Clinical Indication
EGFR	Cetuximab	Erbitux	Chimeric IgG1	Squamous-Cell Carcinoma of the Head and Neck, Colorectal Cancer
	Panitumumab	Vectibix	Human IgG2	Colorectal Cancer
	Necitumumab	Portrazza	Human IgG1	Squamous Non-Small-Cell Lung Cancer
HER2	Trastuzumab	Herceptin	Humanized IgG1	Breast Cancer
	Pertuzumab	Perjeta	Humanized IgG1	Breast Cancer
VEGF	Bevacizumab	Avastin	Humanized IgG1	Colorectal Cancer, Non-Squamous Non-Small Cell Lung Cancer, Glioblastoma, Renal-Cell Carcinoma, Cervical Cancer, Epithelial Ovarian, Fallopian Tube, Primary Peritoneal Cancer, Hepatocellular Carcinoma
VEGFR2	Ramucirumab	Cyramza	Human IgG1	Gastric Or Gastro-Esophageal Junction Adenocarcinoma, Non-Small-Cell Lung Cancer, Colorectal Cancer, Hepatocellular Carcinoma
PD-L1	Atezolizumab	Tecentriq	Humanized IgG1	Non-Small-Cell Lung Cancer, Small-Cell Lung Cancer, Hepatocellular Carcinoma, Melanoma, Alveolar Soft Part Sarcoma
	Avelumab	Bavencio	Human IgG1	Merkel Cell Carcinoma
	Durvalumab	Imfinzi	Human IgG1	Non-Small-Cell Lung Cancer, Small-Cell Lung Cancer, Biliary Tract Cancer, Hepatocellular Carcinoma
GD2	Dinutiximab	Unituxin	Chimeric IgG1	Neuroblastoma
	Dinutiximab-beta	Quarziba	Chimeric IgG1	Neuroblastoma
PDGFRα	Olaratumab	Lartruvo	Human IgG1	Soft-Tissue Sarcoma

**Table 2 cancers-16-01045-t002:** Monoclonal antibodies (mAbs) conjugated to the near-infrared I (NIR-I) fluorescent dye IRDye800CW for the intraoperative imaging of solid extra-hematological malignancies.

Drug	Tumor Type	Phase	Recruitment Status	Estimated Patient Enrollment	Dose	Timing	ClinicalTrials.gov Identifier
Cetuximab-IRDye800CW	Esophageal Cancer	Phase I	nd	40	1% of therapeutic dose (2.5 mg/m^2^); 10% of therapeutic dose (25 mg/m^2^); 25% of therapeutic dose (62.5 mg/m^2^) (iv)	2 days before surgery	NCT04161560
	Head and Neck Cancer	Phase I; Phase II	Recruiting	79	10 mg; 25 mg; 50 mg; 75mg unlabeled test/loading dose cetuximab + 15 mg Cetuximab-IRDye800CW; 75 mg unlabeled test/loading dose cetuximab + 25 mg Cetuximab-IRDye800CW (iv)	4 days before surgery	NCT03134846
	Head and Neck Cancer	Phase I	Terminated	12	10 mg; 100 mg unlabeled test/loading dose of cetuximab + 50 mg dose of cetuximab-IRDye800 (iv)	Prior to surgery	NCT01987375
	Pancreatic Adenocarcinoma	Phase II	Terminated	10	100 mg unlabeled test/loading dose of cetuximab + 50 mg; 100 mg unlabeled test/loading dose of cetuximab + 100 mg (iv)	2–5 days before surgery	NCT02736578
	Brain Neoplasm	Phase I; Phase II	Terminated	10	unlabeled test/loading dose of cetuximab + 50 mg; unlabeled test/loading dose of cetuximab + 100 mg (iv)	2–5 days before surgery	NCT02855086
	Rectal Cancer	Phase I	nd	15	75 mg unlabeled test/loading dose of cetuximab + 15 mg (iv)	Prior to surgery	NCT04638036
	Penile Cancer	Phase I	Recruiting	15	15 mg unlabeled test/loading dose of cetuximab + 15 mg (iv)	2 days before surgery	NCT05376202
	Head and Neck Cancer	Phase II	Not yet recruiting	20	75 mg unlabeled test/loading dose of cetuximab + 15 mg (iv)	Prior to surgery	NCT05499065
Panitumumab-IRDye800CW	Head and Neck Cancer	Phase II	Recruiting	25	50 mg (iv)	nd	NCT04511078
	Head and Neck Cancer	Phase II	Completed	20	50 mg (iv)	1–5 days before surgery	NCT03405142
	Head and Neck Cancer	Phase I	Completed	43	nd (iv)	Prior to surgery	NCT02415881
	Head and Neck Cancer	Phase I	Completed	14	30 mg (iv)	2-5 days before surgery	NCT03733210
	Lung Cancer	Phase I; Phase II	Active, not recruiting	30	nd (iv)	1–2 days or 3–5 days prior to surgery	NCT03582124
	Brain Tumor	Phase I; Phase II	Not yet recruiting	12	0.006 mg/kg; 0.25 mg/kg; 0.5 mg/kg; 1.0 (with max cap dose 50 mg)	1–5 days before surgery	NCT04085887
	Brain Tumor	Phase I; Phase II	Recruiting	22	50 mg; 100 mg unlabeled test/loading dose of panitumumab + 50 mg; 100 mg; 100 mg unlabeled test/loading dose of panitumumab + 100 mg (iv)	1–5 days before surgery	NCT03510208
	Pancreatic Adenocarcinoma	Phase I; Phase II	Active, not recruiting	24	100 mg unlabeled test/loading dose of panitumumab + 25 mg; 100 mg unlabeled test/loading dose of panitumumab + 50 mg; 100 mg unlabeled test/loading dose of panitumumab + 75 mg; 50 mg (iv)	2–5 days before surgery	NCT03384238
Bevacizumab-IRDye800CW	Adenomatous Polyposis Coli	Phase I	Completed	15	4.5 mg; 10 mg; 25 mg (iv)	3 days before endoscopy	NCT02113202
	Rectal Cancer	Phase I	Completed	30	4.5 mg (iv)	2–3 days before endoscopy	NCT01972373
	Breast Cancer	Phase I	Completed	30	4.5 mg (iv)	4 h, 2 days before imaging and 3 days before surgery	NCT01508572
	Breast Cancer	Phase I; Phase II	Completed	26	4.5 mg; 10 mg; 25 mg; 50 mg (iv)	3 days before surgery	NCT02583568
	Soft-Tissue Sarcoma	Phase I; Phase II	Completed	23	10 mg; 25 mg; 50 mg (iv)	Prior to surgery	NCT03913806
	Esophageal Cancer	Phase I	Completed	10	4.5 mg (iv)	2 days before endoscopy	NCT02129933
	Esophageal Cancer	Phase I	Terminated	25	4.5 mg; 10 mg; 25 mg (iv)	Prior to endoscopy	NCT03558724
	Hilar Cholangiocarcinoma	Phase I; Phase II	Recruiting	12	10 mg; 25 mg; 50 mg (iv)	3 days before surgery	NCT03620292
	Barrett Esophagus	Phase II	nd	60	nd (topical)	5 min before endoscopy	NCT03877601
	Inverted Papilloma	Phase I	Active, not recruiting	6	10 mg; 25 mg (iv)	2–4 days before surgery	NCT03925285
	Pancreatic Cancer	Phase I; Phase II	Terminated	26	4.5 mg; 10 mg; 25 mg; 50 mg (iv)	3 days before surgery	NCT02743975
	Pituitary Tumor	Phase I	Recruiting	15	4.5 mg; 10 mg; 25 mg (iv)	Prior to endoscopy	NCT04212793
Girentuximab-IRDye800CW	Renal Tumor	Phase I	Recruiting	22	nd (iv)	7 days before surgery	NCT03558724
Labetuzumab-IRDye800CW	Colorectal Cancer	Phase I; Phase II	Recruiting	29	nd (iv)	6–7 days before surgery	NCT03699332
Nimotuzumab-IRDye800CW	Lung Cancer	Phase I; Phase II	Recruiting	36	50 mg; 100 mg (iv)	4–6 days before surgery	NCT04459065

## Data Availability

The data presented in this study are available in this article.

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
