# Peer review of "Monoclonal Antibodies for Targeted Fluorescence-Guided Surgery: A Review of Applicability across Multiple Solid Tumors"

_cancers, 2024, doi:10.3390/cancers16051045_

Round 1
Reviewer 1 Report
Comments and Suggestions for Authors
The manuscript cancers-2861258 with the title "Monoclonal Antibodies for Targeted Fluorescence-Guided Surgery: A Review of Applicability across Multiple Solid Tumors" summarized mAbs for the development of targeted fluorescence-guided surgery. Although the theme is interesting, the discussion are insufficient. Some comments:
1. The format of this manuscript should be modified to easy understand for author.
2. Although different mAb-fluorophore conjugates were summarized, their advantages, disadvantages, and bottleneck should be pointed out. Some efficient methods should be suggested for these works.
3. Discussion: to evolve the development of targeted fluorescence-guided surgery, the development trends of mAb-fluorophore conjugates should be discussed.
4. Comparison between other guided surgery and mAb-fluorophore conjugates guided surgery should be added.
5. Some key figures and the structures of typical mAb-fluorophore conjugates should be added.
Comments on the Quality of English LanguageNo
Author Response
Dear Reviewer,
Our group would like to thank you for the opportunity to resubmit a revised version of our manuscript.
Attached you can find a thoroughly revised version of the manuscript with the highlighted changes from the original version.
- The format of this manuscript should be modified to easy understand for author.
The format of the manuscript has been chosen to focus on the utilization of monoclonal antibodies (mAbs) in targeted fluorescence-guided surgery (T-FGS), particularly focusing on mAbs conjugated with the dye IRDye800CW. We believe we manage to be succinct and focused in preparing a technical manuscript that can deliver important information to scientists and clinicians willing to translate T-FGS. Notably, we have also provided two tables and ClinicalTrials.gov Identifiers for approximately 31 clinical trials employing mAbs-IRDye800CW. This inclusion was aimed to facilitate ease of access for researchers seeking detailed information from a reliable source. Please let us know your thoughts if you believe there is a better way to organize the paper.
- Although different mAb-fluorophore conjugates were summarized, their advantages, disadvantages, and bottleneck should be pointed out. Some efficient methods should be suggested for these works.
This review paper focuses on presenting clinical evidence on the use of mAbs as the method to deliver targeted fluorescence-guided surgery (T-FGS) in adult and pediatric solid cancers. The utility, indications, doses, and timing of administration of the most promising mAbs to be used for T-FGS in oncology patients are presented to support clinical translation. Limits, disadvantages, and bottlenecks of the current applications of T-FGS are presented in the discussion.
- Discussion: to evolve the development of targeted fluorescence-guided surgery, the development trends of mAb-fluorophore conjugates should be discussed.
Thanks for this important comment. To facilitate the development of targeted fluorescence-guided surgery (T-FGS), we have divided the manuscript into three different sections: i) monoclonal antibodies (mAbs) clinically approved for the treatment of extra-hematological solid malignancies under evaluation in clinical trials for T-FGS purposes; ii) monoclonal antibodies (mAbs) clinically approved for the treatment of extra-hematological solid malignancies not yet under evaluation in clinical trials for T-FGS purposes; iii) promising monoclonal antibodies (mAbs) for T-FGS purposes not yet clinically approved for the treatment of extra-hematological solid tumors. In the discussion, possible evolution and future improvements in the field were highlighted including working in the NIR-II, NIR-III, and NIR-IV spectrum, adopting a new generation of small molecules (i.e. affibodies and nanobodies), and using intraoperative sprayable targeted probes to overcome the issue linked to the quick quenching process.
- Comparison between other guided surgery and mAb-fluorophore conjugates guided surgery should be added.
Thanks for your comment. We believe the paper is already long and full of information and for this reason, we kept the discussion focused. The manuscript aimed to review the status of the clinical use of monoclonal antibodies (mAbs) that have completed or are in ongoing clinical trials for targeted fluorescence-guided surgery (T-FGS) for the intraoperative identification of the tumor margins of solid tumors to speed up the translation to the operating room. In addition, in the discussion, we have added the role and possible value of fluorophores working in the NIR-II, NIR-III, NIR IV spectrum, small molecules (i.e. affibodies and nanobodies) that could be adopted as theranostic tools to provide high-contrast images within a few hours after injection (overcoming the slow and low tumor uptake and the limited tumor penetration compared to traditional mAbs) and sprayable targeted probes that can make their clinical use simpler, more convenient, and possibly with less systemic side effects. More information on other guided surgery techniques will be out of place and will make the paper too long and heavy to read.
- Some key figures and the structures of typical mAb-fluorophore conjugates should be added.
We appreciate your comment and, as suggested, we have now added a Figure showing the conjugation of anti-GD2 monoclonal Antibodies (mAbs) with IRDye800CW in-vitro and in-vivo experiments.
We thank you all very much for the useful comments, which have certainly improved the quality of our manuscript.
We look forward to your comments and suggestions regarding our re-submission.
Yours sincerely.
Reviewer 2 Report
Comments and Suggestions for Authors
The manuscript centers on the utilization of monoclonal antibodies (mAbs) in targeted fluorescence-guided surgery, particularly focusing on mAbs conjugated with the dye IRDye800CW. The authors offer a succinct analysis of the current landscape of clinical trials involving this application.
The strength of the manuscript lies in its brevity, devoid of unnecessary details. Notably, the authors provide ClinicalTrials.gov Identifiers for approximately 31 clinical trials employing mAbs-IRDye800CW. This inclusion facilitates ease of access for researchers seeking detailed information from a reliable source.
Author Response
Dear Reviewer,
Our group would like to thank you for your kind comments.
Yours sincerely.
Reviewer 3 Report
Comments and Suggestions for Authors
In this comprehensive review, the authors present the present status of clinical use of monoclonal antibodies equipped with tracers for use in targeted fluorescence guided surgery, which aids intraoperative identification of extra-hematological solid tumors. In many cases, surgical resection of tumor tissue is still vital for effective treatment, and the method presented supports an improved tumor localization by image-guided surgery, rather than having to rely on palpation and visual inspection. Importantly, the perspectives of the method as well as its limitations (previous knowledge on tumor-surface expressed antigens) are also discussed. Please find below a list of minor remarks which I hope you will find helpful.
Line 159: labeled with fluorophores
Line 162: 25.0 and 62.5 should be the switched
Line 189: “100 mg of unlabelled Panitumumab patients were administered” – please correct
Line 199: mg/kg
Line 254: 4.5 mg?
Line 339: Should this be a subsection title?
Line 348: “…an anti-EGFR recombinant human mAb…that binds to human EGFR”, antigen repeats itself plus it is necitumumab, not necituximab
Line 357: Trastuzumab is also the basics of 2 approved antibody-drug conjugates (T-DM1 and trastuzumab-deruxtecan), which is surely worth mentioning here.
Line 449: I would propose to move this paragraph next to the anti-VEGFR antibodies (synergism in countering VEGF-mediated activity)?
Line 463: subsection title?
Line 531: please include a reference for this promising approach
Supplementary material: are these the same Tables as in the main text?
Author Response
Dear Reviewer,
Our group would like to thank you for the opportunity to resubmit a revised version of our manuscript.
Attached you can find a thoroughly revised version of the manuscript with the highlighted changes from the original version.
We thank you all very much for the useful comments, which have certainly improved the quality of our manuscript, and we hope it will now be acceptable for publication in Cancers.
We look forward to your comments and suggestions regarding our re-submission.
Yours sincerely.